# Slimmed Asymmetrical Contrastive Learning and Cross Distillation for Lightweight Model Training

**Jian Meng,**[*] **Li Yang,**[†] **Kyungmin Lee,**[‡] **Jinwoo Shin,**[‡] **Deliang Fan,**[§] **and Jae-sun Seo**[*]

[*]Cornell Tech, USA      [†]University of North Carolina at Charlotte, USA

[‡]KAIST, South Korea     John Hopkins University, USA

`{kyungmnlee, jinwoos}@kaist.ac.kr`[‡] `lyang50@uncc.edu`[†]

`dfan10@jhu.edu`[§]   `{jm2787, js3528}@cornell.edu`[*]

## Abstract

Contrastive learning (CL) has been widely investigated with various learning mechanisms and achieves strong capability in learning representations of data in a self-supervised manner using unlabeled data. A common fashion of contrastive learning on this line is employing large-sized encoders to achieve comparable performance as the supervised learning counterpart. Despite the success of the labelless training, current contrastive learning algorithms *failed* to achieve good performance with lightweight (compact) models, e.g., MobileNet, while the requirements of the heavy encoders impede the energy-efficient computation, especially for resource-constrained AI applications. Motivated by this, we propose a new self-supervised CL scheme, named SACL-XD, consisting of two technical components, **S**limmed **A**symmetrical **C**ontrastive **L**earning (SACL) and **Cross-D**istillation (XD), which collectively enable efficient CL with compact models. While relevant prior works employed a strong pre-trained model as the teacher of unsupervised knowledge distillation to a lightweight encoder, our proposed method trains CL models from scratch and outperforms them even without such an expensive requirement. Compared to the SoTA lightweight CL training (distillation) algorithms, SACL-XD achieves 1.79% ImageNet-1K accuracy improvement on MobileNet-V3 with $64\times$ training FLOPs reduction. Code is available at `https://github.com/mengjian0502/SACL-XD`.

## 1 Introduction

To overcome the labeling bottleneck for supervised training of deep neural networks (DNNs), self-supervised learning has been widely investigated to learn representations without intensive labeling. In particular, contrastive learning (CL) has demonstrated its capability of representation learning in various machine learning domains, e.g., image [7, 34], video [31], language [2], speech [11], and medical imaging [30]. The success of CL is built upon different data augmentations from the original (training) samples, and the representation is learned by maximizing the latent common knowledge between contrastive embeddings [8, 7, 35], which are separately encoded from the augmented images by DNN models [25]. Despite the various contrastive learning techniques, learning the latent knowledge and representations requires wide and deep encoders. In particular, current SoTA CL algorithms [3, 18] have to employ a large-sized encoder (e.g., ResNet-50) to achieve comparable performance as the supervised learning counterpart.

On the other hand, training the lightweight models [23, 22, 29] from scratch is largely under-explored in CL. Almost none of the prior works have reported the CL performance with directly-trained lightweight models. In fact, the prior success of the lightweight models in supervised learning *cannot* be smoothly transferred to CL. where the performance gap between the lightweight models and large-sized models is largely amplified under recent CL methods [7, 19], For example, with

supervised learning, MobileNet-V3Large (Mob-V3) [22] can achieve 74.04% on ImageNet-1K, which is comparable to ResNet-50 (76.15%). However, the ∼2% accuracy difference is amplified to >30% (75.2% → 36.3%) in CL, as reported in [16] with MoCo-V2. Training MobileNet-V3 with more recent Barlow Twins [35] improves the accuracy to ∼52%, but this is still unsatisfactory.

To overcome the limited trainability of the lightweight DNN models in CL, contrastive lightweight model learning has been investigated as an unsupervised knowledge distillation task. For example, SEED [16] divides the entire training process into a two-step process of (1) teacher pre-training with CL and (2) unsupervised knowledge distillation from the frozen teacher to the student lightweight encoder (e.g., MobileNet-V3 [22]). In addition to the two-step process of "pretraining-and-tuning", the authors employ different input schemes between two steps, which further elevates the complexity of the entire training process. On the other hand, ReKD [36] introduces the latent relation knowledge into online distillation with a single-step training process. However, the large-sized encoder is still required as the online teacher.

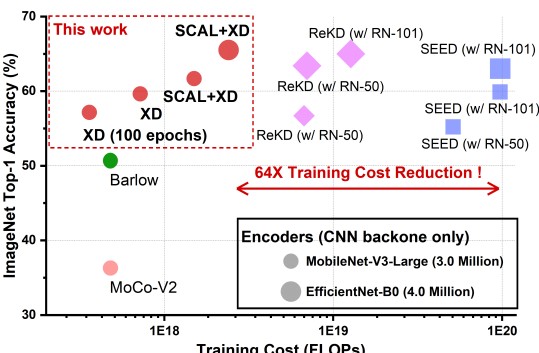

Figure 1: ImageNet-1K accuracy of the proposed method (SACL+XD) vs. state-of-the-art methods. "RN" represents the ResNet teacher of the prior works [16, 36]. Following the settings in [36, 16], the number of FLOPs is computed based on 200 epochs of training.

To that end, both ReKD [36] and SEED [16] focus on unsupervised distillation with the large-sized teacher, which actually amplifies the training cost for lightweight CL, as shown in Fig. 1. Meanwhile, the performance of training lightweight models with CL from scratch remains unsatisfactory [27] or requires dedicated input scaling design with low generality [28] on normal CL. This is quite meaningful to explore since training a high-performance compact encoder (≤ 5 M parameters) [22, 23, 29] by necessitating a large-sized ResNet (>20 M) [20] teacher largely degrades the "time-to-deploy" of the model due to the magnified training cost. Given the challenges of CL and the limitations of prior works, we raise the following question as our motivation:

*Is there a contrastive learning algorithm that can train the high-performance lightweight model without using a large-sized teacher?*

To answer this question, we propose **S**limmed **A**symmetrical **C**ontrastive **L**earning (SACL) together with **C**ross-**D**istillation (XD), a novel self-supervised contrastive learning algorithm designed for efficient and high-performance CL with lightweight encoders. Specifically, SACL considers lightweight contrastive learning as an asymmetrical sparse training problem based on the shared weights from a dense host encoder. A lightweight encoder can be treated as a "subset" model (sub-model) sliced from a wide host model, which naturally formulates the asymmetrical encoding in CL. Different from the knowledge distillation-based methods [16, 36], the proposed SACL-XD algorithm completely eliminates the large-sized ResNet teacher from the lightweight contrastive learning. On top of that, we introduce cross-distillation (XD) to facilitate SACL. XD minimizes the decorrelation between the latent information distorted by asymmetrical CL, elevating the training stability and performance. As shown in Fig. 1, the proposed method achieves **64×** and **8.3×** training cost reduction along with **1.79**% and **2.09**% ImageNet accuracy improvements, compared to SOTA [16] and [36], respectively. The major contributions of the proposed SACL-XD algorithm are:

1. **Simplicity:** SACL-XD does not require teacher pre-training (and input scaling) before training a lightweight student encoder.
2. **Efficiency:** SACL-XD does not require the large-sized teacher during training. The training cost of MobileNet-V3 [22] model is reduced by 64× and 8.3× compared to [36, 16].
3. **Performance and generality:** SACL-XD achieves up to 2.09% ImageNet-1K accuracy improvement compared to the previous SoTA method. In particular, the proposed cross-distillation (XD) algorithm *solely* achieves superior performance with normal contrastive learning settings on ResNet-50 without SACL.
4. **Transferability:** The SACL-XD-trained lightweight encoder shows high transferability to the downstream tasks. With minimum fine-tuning on top of the ImageNet-1K-trained

MobileNet-V3, ours achieves **94.80%** accuracy on CIFAR-10, outperforms the supervised learning baseline and SEED [16] with **1.83%** and **14.80%** improvements, respectively.

## 2    Related Work

**Self-supervised contrastive learning.** With the absence of deterministic labels, the major objective of contrastive learning is minimizing the distance between the embeddings separately encoded from the augmented input samples. Early research works [7, 19] define "positive" and "negative" sample pairs, and the learning process maximizes the similarity between positives and repels the negative samples. The popular InfoNCE loss [26, 19, 8] or NT-Xent loss [7] has been proposed as the learning objective. The wide and deep DNN encoder is also the key factor of success in contrastive learning. In addition to the cross-reference between positive and negative pairs, contrastive learning is also considered a DNN-based clustering problem, where the samples and their embeddings are grouped into clusters based on similarity metrics [4, 32, 1]. SwAV [5] introduces online cluster learning with the reduced complexity of computation. The entropy-based similarity matching and clustering CL algorithms shares the same nature, but they all require an extensive amount of negative samples to generate salient contrastiveness.

Recent methods consider contrastive learning as a knowledge distillation [21] task between the contrastive encoders with separately augmented inputs. BYOL [18] and SimSiam [9] consider the student network as the online model, while the teacher is consistently staying offline with no gradient propagation [9]. The teacher is updated as the moving averaged weights of the student encoder. The "student-teacher" relationship focuses on the distance minimization between latent information. Another perspective is exploring the content hidden inside the embeddings. Barlow Twins [35] pushes the cross-correlation between the encoded embeddings toward the identity matrix. VICReg [3] collectively optimizes the variance, covariance, and distance between and inside the embeddings to achieve better training stability and even enables the CL with different encoders. Among all the previous CL algorithms, the empirical findings show the improved training stability and performance of CL together with the relaxed requirement of batch sizes, but the large-sized encoder (e,g., ResNet-50 ($1\times, 2\times, \ldots$) [20]) is a persistent and almost mandatory requirement to achieve high performance, which hinders the development of CL with high energy consumption and extensive training effort.

**Contrastive learning with lightweight models.** Under the context of supervised learning, various lightweight architecture designs [23, 22, 29] are proposed to minimize the performance difference caused by the model size gap. The intricate architecture and the compact model sizes enable energy-efficient and hardware-friendly computer vision applications. However, the architectural efficiency of the lightweight models becomes invalid in the contrastive learning domain. Employing a lightweight encoder in CL leads to poor performance due to the insufficient trainability caused by the limited model sizes. To resolve this challenge, the mainstream method is introducing a giant model as the teacher of the lightweight student encoder. SEED [16] uses a pre-trained large-sized ResNet [20] as the frozen teacher of the lightweight student model. The similarity-based teacher distribution is generated as the target soft label for student learning. In particular, accurate distillation requires both teacher and student to encode the same input, which is inconsistent with the augmentation strategy of CL. As a result, self-supervised training of the lightweight model becomes a two-step process. However, the dependence on the large-sized teacher and the expensive pre-training complicates the overall learning process. ReKD [36] incorporates the online distillation during learning with the relation knowledge-based loss between embeddings. DisCo [17] complicates the training cost of the distillation even further. The inputs are encoded separately by the teacher and *two* separate students. Regardless of using an online or frozen teacher, the large-sized ResNet model is always required to train a lightweight encoder. The recent efforts on directly-trained lightweight model requires fine-grained efforts on input scaling [28], but the improvements on lightweight model has low generality and has **no** improvements on the popular ResNet-50-based CL.

## 3    Method

To resolve the training dilemma between contrastive learning and lightweight models, this section introduces 1) SACL: Slimmed Asymmetrical Contrastive Learning and 2) XD: Cross Distillation to achieve lightweight contrastive learning from scratch.

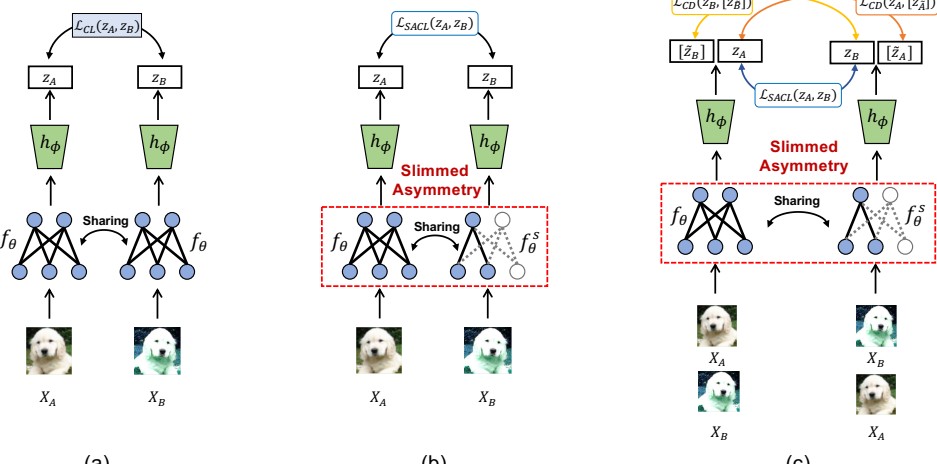

Figure 2: Overview of (a) contrastive learning with shared encoder, (b) Proposed Slimmed Contrastive Learning (SACL) algorithm and (c) Cross-Distillation-aided SACL.

Similar to prior contrastive learning (CL) algorithms [35, 3, 7], our method performs contrastive learning based on the dual augmentation that is transformed from the clean input image. Given a clean input batch $X$ sampled from the dataset $\mathcal{D}$, the distorted views are generated from a combination of data augmentations $\mathcal{T}$, leading to the augmented pairs $X_A$ and $X_B$, which are fed into the encoders for different contrastive paths. In this work, we follow the recent contrastive learning algorithms [35, 3] and use one shared encoder $f_\theta$ for different contrastive branches, as shown in Fig. 2(a). The encoded outputs are fed into the subsequent projector $h_\phi$, resulting in the latent embeddings $z_A$ and $z_B$.

### 3.1 Slimmed Asymmetric Contrastive Learning (SACL)

Under the standard supervised training, it has been well evidenced that extracting a subset lightweight model out of a wider/larger (e.g., $2\times$) encoder often leads to better performance compared to directly train such a lightweight model from scratch [37]. We are inspired by the fact that a lightweight model can be considered as a subset model "sliced" from the original full-sized host model. The asymmetrical relationship between the slimmed model and the original model naturally fits the independent encoding path of contrastive learning.

We propose Slimmed Asymmetric Contrastive Learning (SACL), for obtaining lightweight models under self-supervised CL training. SACL treats the lightweight model contrastive learning as a sparse training optimization problem based on shared weights from a large-sized encoder. During the forward pass of each iteration, the augmented input pair $(X^A, X^B)$ are separately encoded by the dense host model ($f_\theta$) and the slimmed ($f_\theta^s$) encoders with the disabled weight filters (i.e., output channels), as shown in Fig. 2(b).

The host encoder model $f_\theta$ is slimmed by removing a *unified* amount of weight filters (output channels) of each layer based on weight magnitude score. With the slimmed and dense model $f_\theta^s$ and $f_\theta$, the relationship between $\theta^s$ and $\theta$ is defined as:

$$\theta_s \subset \theta \quad \text{and } \theta_s = \theta \cdot \mathcal{M} \tag{1}$$

Where $\mathcal{M}$ is the weight `mask` that disables the channels of $f_\theta$, and the subset model $f_\theta^s$ is selected when $\mathcal{M}$ is enabled. As shown in Fig. 2(b), $f_\theta$ and $f_\theta^s$ separately encode the input pair in different branches. With the shared weights, the mask $\mathcal{M}$ is alternatively enabled between branches to formulate the asymmetrical encoding during the forward propagation. The slimming ratio (sparsity) of $\mathcal{M}$ is defined by the *Slimmed Asymmetry* (SA) between $f_\theta$ and $f_\theta^s$ with the style of "K×-1×". "K" is the width of the wide host model ($f_\theta$) that is employed to slice the $1\times$ model ($f_\theta^s$) out of it:

$$s = 1 - 1/\text{K} = \text{card}(\theta_s)/\text{card}(\theta) \tag{2}$$

where card($\cdot$) returns the number of nonzero elements of the tensor, $s$ is the desired slimmed ratio (sparsity) controlled by K.

Starting from the initialization of the training, SACL uniformly generates the masks to slice out the weight filters (output channels) with the least magnitude score from each layer. The masks will be

updated after each epoch to maintain the performance of the slimmed model based on the filter-wise $L_1$ norm score. The resultant slimmed ($1\times$) model will be deployed as the final trained encoder. Specifically, SACL drives contrastive learning with the following properties:

1. SACL holds a consistent channel-wise architecture difference (e.g., $1.5\times$ vs. $1\times$) between contrastive branches throughout the entire training process.

2. SACL removes a unified amount of channels (with the lowest magnitude score) of all the layers. By doing so, the resultant slimmed model will have the exact width as the target lightweight model.

Formally, given the augmented input pair $(X_A, X_B)$, the forward pass is characterized as:

$$z_A = h_\phi(f_\theta(X_A)), \qquad z_B = h_\phi(f_\theta^s(X_B)). \tag{3}$$

And the optimization target of SACL is:

$$\min_{\theta,\theta_s} \mathcal{L}_{\text{SACL}}(z_A, z_B), \qquad \text{such that} \quad \theta_s \subset \theta \text{ and } \frac{\text{card}(\theta_s)}{\text{card}(\theta)} = s, \tag{4}$$

Inside each mini-batch, the gradient is collectively computed all at once, and the optimizer will update the whole set parameter $\theta$ of the dense host encoder $f_\theta$. The slimmed model architecture ($f_\theta^s$) will be updated after every epoch based on the magnitude score of each filter. The model is completely online and stop gradient is excluded from learning. Minimizing the contrastive loss $\mathcal{L}_{\text{SACL}}$ between $z_A$ and $z_B$ is equivalent to overcoming the distortion caused by 1) data augmentation $\mathcal{T}$ and 2) consistent and structural architecture difference between $f_\theta^s$ and $f_\theta$. And the architecture asymmetry caused by 2) motivates us to explore the enhancement of SACL from the perspective of knowledge distillation, which is presented in the following section.

## 3.2  Cross Distillation (XD) on top of SACL

We propose Cross-Distillation (XD) on top of SACL, which treats the teacher-student relationship as an interconnected knowledge distillation with the correlation-based optimization on top of the proposed SACL learning scheme. Given the asymmetrical contrastive encoders $f_\theta$ and $f_\theta^s$, we first encode $X^A$ and $X^B$ based on SACL ($X^A \to f_\theta$; $X^B \to f_\theta^s$), leading to the embeddings $z^A$ and $z^B$. Subsequently, we freeze both $f_\theta$ and $f_\theta^s$ while **reversing** the order of the inputs for encoding ($X^B \to [f_\theta] \to [\tilde{z}_B]$; $X^A \to [f_\theta^s] \to [\tilde{z}_A]$), and characterize the resultant embeddings as $[\tilde{z}_A]$ and $[\tilde{z}_B]$, where "$[\cdot]$" represents the frozen encoder for the forward pass only.

As a result, each forward pass will generate two pairs of latent vectors resulting from two groups of SACL forward pass. We first compute the online contrastive loss $\mathcal{L}_{\text{SACL}}$ based on $z_A$ and $z_B$. Such online embeddings contain the distortion caused by *both* data augmentations and asymmetrical encoders. We empirically find out that directly optimizing such high-sparsity difference via single contrastive loss leads to *collapsed* training. Motivated by that, we compute the cross-distillation loss $\mathcal{L}_{\text{CD}}$ as the average loss between the pair of $(z_A, \tilde{z}_A)$ and $(z_B, \tilde{z}_B)$:

$$\mathcal{L}_{\text{CD}} = \frac{\mathcal{L}_{\text{CD}}^A(z_A, \tilde{z}_A) + \mathcal{L}_{\text{CD}}^B(z_B, \tilde{z}_B)}{2}, \tag{5}$$

where $\mathcal{L}_{\text{CD}}^A, \mathcal{L}_{\text{CD}}^B$ will be defined formally later (see Eq. 9). As shown in Fig. 2, each term of the cross-distillation loss is computed between the asymmetrical SACL encoders with the same input. In other words, optimizing the cross-distillation loss is equivalent to minimizing the distortion in embeddings caused by the asymmetrical sparsity only. We define the total loss of the training as the weighted sum between $\mathcal{L}_{\text{SACL}}$ and $\mathcal{L}_{\text{CD}}$ and weight is defined as $\alpha$:

$$\mathcal{L} = \alpha\mathcal{L}_{\text{SACL}} + (1 - \alpha)\mathcal{L}_{\text{CD}}, \tag{6}$$

Where $n$ represents the index of batch, and $i$ and $j$ represent the dimensionality indices across the latent output. Correspondingly, $C_{i,j}$ represents the $i, j$ element of the correlation matrix.

Regarding contrastive loss design, recent works treat contrastive learning as the optimization problem of principle correlation maximization [35] or the decorrelation minimization [35, 3] of the encoded latent vectors. The cross-correlation between $z_A$ and $z_B$ is computed as:

$$C_{ij} = \frac{\sum_n z_{n,i}^A z_{n,j}^B}{\sqrt{\sum_n (z_{n,i}^A)^2}\sqrt{\sum_n (z_{n,j}^B)^2}} \tag{7}$$

Table 1: ImageNet-1k test accuracy with linear evaluation protocal based on MobileNet-V3 [22] trained by different contrastive learning/distillation methods.

| Method | Linear Eval. Acc. (%) | Training Epochs | Pre-training |
|---|---|---|---|
| **XD. (Ours)** | **57.16** | 100 | ✗ |
| ReKD [36] | 56.70 | 200 | ✗ |
| SEED [16] | 55.20 | 200 | ✓ |

In this work, we first adopt the BT-loss [35] as the starting point of the online contrastive loss $\mathcal{L}_{\text{SACL}}(z_A, z_B)$, where the principle correlation (diagonal) is maximized, and the cross-decorrelation (off-diagonal) is minimized during training.

$$\mathcal{L}_{\text{SACL}}(z_A, z_B) = \sum_i (1 - C_{ii}^{AB}) + \lambda \sum_i \sum_{i \neq j} (C_{ij}^{AB})^2 \tag{8}$$

Unlike the prior work that uses the SoftMax-based probability [16] as unsupervised soft labels for distillation, we find out the proposed cross-distillation task between $z$ and $\tilde{z}$ can still be considered as a correlation-based optimization problem, **and our finding shows the proposed cross-distillation is a strong facilitator for contrastive learning even without SACL.** First, by duplicating $L_{\text{SACL}}$ for $L_{\text{CD}}$, each term in Eq. 5 becomes:

$$\mathcal{L}_{\text{CD}}^A = \underbrace{\sum_i (1 - C_{ii}^{A\tilde{A}})}_{\text{inner-correlation loss}} + \lambda \underbrace{\sum_i \sum_{i \neq j} (C_{ij}^{A\tilde{A}})^2}_{\text{inner-decorrelation}} \tag{9}$$

By encoding the same input with the identical dense model, the *inner correlation* $C_{ii}^{A\tilde{A}} \to 1$, and *inner de-correlation* $\sum_i (1 - C_{ii}^{A\tilde{A}}) \to 0$, which makes the cross-distillation loss $\mathcal{L}_{\text{CD}}^A$ and $\mathcal{L}_{\text{CD}}^B$ equivalent to the internal decorrelation loss between different dimensions inside $z_A$ and $z_B$, respectively. As a result, minimizing $\mathcal{L}_{\text{CD}}$ avoids the aliasing feature across different dimensions. We would like to highlight the fact that the combination in Eq. 10 with both cross-decorrelation ($L_{\text{SACL}}$) and the inner decorrelation ($L_{\text{CD}}$) is important for contrastive learning and leads to superior performance compared to [3, 35]. Table 1 summarizes the linear evaluation accuracy of the lightweight MobileNet-V3-Large [22] trained on the ImageNet-1K dataset. Starting from scratch with only 100 epochs, the proposed Cross-distillation (XD) method achieves 2% higher accuracy compared to [16], which uses ResNet-50 ($1\times$) as the teacher. In Section 4, we also compare the proposed XD algorithm with other CL methods on ViT [13] and ResNet-50 [20] without SACL. The superior performance of XD proves the importance of the co-optimization between cross-decorrelation ($L_{\text{SACL}}$) and the inner decorrelation ($L_{\text{CD}}$).

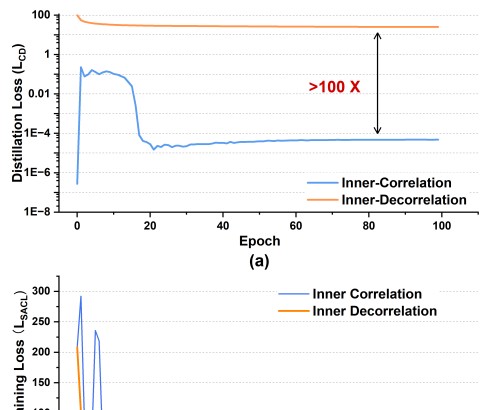

Figure 3: (a) Negligible magnitude of inner correlation with the SACL MobileNet-V1 ($1.5\times$-$1\times$) trained on ImageNet-1K. (b) Unstable training and the degraded performance caused by the $L_{\text{CD}}$ with inner correlation only.

Combining the cross-distillation with SACL, the *inner decorrelation* between $z$ and $[\tilde{z}]$ plays the dominant role in the distillation loss $\mathcal{L}_{\text{CD}}$, and we observe that it is also critical to the training stability and performance of SACL. As shown in Fig. 3(a), the magnitude of the inner correlation is negligible compared to the inner decorrelation, even with the asymmetric encoding of SACL. Furthermore, solely minimizing the infinitesimal inner correlation as the $\mathcal{L}_{\text{CD}}$ introduces instability and degraded performance to the training process, as shown in Fig. 3(b). As a result, we formulate the total loss as:

$$\mathcal{L} = \alpha \mathcal{L}_{\text{SACL}} + (1 - \alpha) \big( \lambda \sum_i \sum_{i \neq j} (C_{ij}^{A\tilde{A}})^2 + \lambda \sum_i \sum_{i \neq j} (C_{ij}^{B\tilde{B}})^2 \big) \tag{10}$$

The detailed pseudocode of the proposed method is summarized in the Appendix A.1. We evaluate the impact of the weight $\alpha$ in Section 4.4. We also investigate the conventional `negative-logarithm` distillation loss in Appendix A.2.

Table 2: ImageNet-1K test accuracy with linear evaluation protocal based on MobileNet-V3 [22] trained by different contrastive learning/distillation methods.

| Method | Encoder | Linear Eval. (%) | Epochs | Pre-train | Teacher | Training FLOPs (e+17) |
|---|---|---|---|---|---|---|
| [‡]SACL-XD (Ours) | Eff-B0 (1.5×-1×) | **65.32 (+2.12)** | 200 | ✗ | - | **24 (2.9× ↓)** |
| [§]SACL-XD (Ours) | Mob-V3 (1.5×-1×) | **61.69 (+1.79)** | 200 | ✗ | - | **15 (64.7× ↓)** |
| **SACL-XD (Ours)** | Mob-V1 (1.5×-1×) | **59.34** | 200 | ✗ | - | **19** |
| **XD only (Ours)** | Mob-V3 (1×) | **59.42** | 200 | ✗ | - | **7.2** |
| **XD only (Ours)** | Mob-V3 (1×) | **57.16** | **100** | ✗ | - | **3.6** |
| **XD only (Ours)** | Mob-V1 (1×) | **55.84** | **100** | ✗ | - | **9.0** |
| [§]SSL-Small [27] | Mob-V3 (1×) | 48.70 | 200 | 2 epochs | - | 19 |
| [§]SSL-Small [27] | Eff-B0 (1×) | 55.90 | 200 | 2 epochs | - | 34 |
| ReKD [36] | Mob-V3 (1×) | 56.70 | 200 | ✗ | ResNet-50 | 67 |
| ReKD [36] | Mob-V3 (1×) | 59.60 | 200 | ✗ | ResNet-101 | 125 |
| ReKD [36] | Eff-B0 (1×) | 63.40 | 200 | ✗ | ResNet-50 | 70 |
| OSS [10] | Eff-B0 (1×) | 64.10 | 800+200 | ✗ | ResNet-50 | 67 |
| [*]SEED [16] | Mob-V3 (1×) | 55.20 | 800+200 | ✓ | ResNet-50 | 512 |
| [*]SEED [16] | Mob-V3 (1×) | 59.90 | 800+200 | ✓ | ResNet-101 | 971 |
| [*]SEED [16] | Eff-B0 (1×) | 61.30 | 800+200 | ✓ | ResNet-50 | 516 |
| [†]MoCo-V2 [8] | Mob-V3 (1×) | 36.30 | 200 | ✗ | - | 4.8 |
| [†]MoCo-V2 [8] | Eff-B0 (1×) | 42.20 | 200 | ✗ | - | 8.5 |

[*]: SEED [16] uses a ResNet-50 teacher which is pre-trained by 800 epochs.
[†]: Baseline linear evaluation accuracy reported by SEED [16].
[‡]:We use ReKD [36] as the SOTA baseline of EfficientNetB0 [29] to report the accuracy improvements and computation reduction.
[§]:We use SEED [16] as the SOTA baseline of MobileNet-V3 [22] to report the accuracy improvements and computation reduction.
[§]:Weights are initialized based on SEED [16]-trained model.

## 4 Experimental Results

In this section, we evaluate the performance of the proposed algorithm based on CNN encoders (MobileNet [23, 22], EfficientNet [29], ResNet [20]) and ViT [13] models on the ImageNet-1K and ImageNet-100 dataset. We also demonstrate the capability of the proposed method with tiny-sized ResNet on the small CIFAR dataset. We also evaluate the transferability of the lightweight model on both CIFAR classification and VOC object detection downstream tasks. We characterize the asymmetry of SACL with the style of "K×-1×", where "K" is the width of the wide host model that is employed to slice the 1× out of it. All the models are directly trained from scratch. The detailed experiment setup and hyperparameter settings are summarized in the Appendix.

### 4.1 Training from Scratch with SACL+XD on Lightweight CNNs

We follow the linear evaluation protocol on ImageNet to evaluate the performance of the backbone trained by the proposed SACL and cross-distillation (XD) algorithm. We train the compact models from scratch for 100 or 200 epochs, which is the same amount of fine-tuning effort as SEED [16] and ReKD [36]. The proposed algorithm is evaluated on multiple lightweight encoders, including MobileNet-V1 [23], MobileNet-V3-Large [22], and EfficientNet-B0 [29]. Table 2 compares the top-1 linear evaluation accuracy of our work against recent SoTA works for compact model training.

With the same 200 epochs, the XD-trained MobileNet-V3 (1×) model outperforms the recent ResNet-50-aided ReKD [36] by a noticeable 2.72% accuracy improvements (59.42% vs. 56.70%) with 9.3× less training FLOPs. Furthermore, XD achieves 57.16% ImageNet accuracy with only 100 epochs training on MobileNet-V3, surpasses the ResNet-50-aided ReKD [36] and SEED [16] by 0.46% and 1.96% with 18.6× and 142× training cost reduction, respectively. Combined with the proposed asymmetrical slimmable contrastive learning (SACL), our method achieves 2.12% and 2.09% linear evaluation accuracy improvements compared to the ReKD-trained EfficientNetB0 and MobileNet-V3, respectively. Meanwhile, our method eliminates the ResNet-101 teacher from training, which leads to 5.2× and 8.3× training effort reduction compared to ReKD [36]. We are aware the wider model introduces a higher computation budget, so we choose the 1.5×-1× asymmetrical architecture, and the linear evaluation is performed on the slimmed encoder with the same size as the vanilla 1× model.

## 4.2 Training from Scratch with XD on ResNet and ViT

Besides the evaluation against the lightweight models, we validate the proposed XD individually by training ResNet-50 on ImageNet-1K with 300 epochs from scratch, as reported in Table 3.

Table 3: ImageNet-1K test accuracy with linear evaluation protocol based on ResNet-50 encoder.

| Method | Training Epochs | Top-1 Linear Evaluation Accuracy (%) |
|---|---|---|
| MoCo [19] | 1000 | 60.6 |
| SimCLR [7] | 1000 | 69.3 |
| *BYOL [18] | 300 | 68.4 |
| *Barlow Twins [35] | 300 | 70.7 |
| **XD (Ours)** | 300 | **71.1** |

\*: Reported results from [18, 35] with 300 epochs training from scratch.

Compared to the recent contrastive learning methods [35, 18], the proposed cross-distillation algorithm (XD) achieves better accuracy with 0.4% linear evaluation accuracy improvements demonstrating the generality and versatility of the proposed XD algorithm.

Besides the CNN-based encoder, the proposed cross-distillation (XD) algorithm is also capable of training the lightweight ViT encoder. Table 4 summarizes the performance of the ViT-Tiny-16-224 [13] encoder trained by XD. Compared to Barlow Twins [9] and DINO [9], our method achieves 1.36% and 0.88% accuracy improvements. The model is directly trained from scratch, and the $\alpha$ value of the XD is set to 0.8. Specifically, we simply replace the CNN encoder by the ViT-Tiny model and slim down the embedding dimensionality, no additional architecture has been introduced to ViT training. The superior performance on ViT indicates the high versatility of the proposed method.

Table 4: ImageNet-100 test accuracy with linear evaluation protocol based on ViT-Tiny [13] encoder.

| Methods | Encoder | Training Epochs | Linear Eval Acc. (%) |
|---|---|---|---|
| Barlow Twins [35] | ViT-Tiny [13] | 400 | 62.56 |
| *DINO [9] | (# of Param = 5.5 Million) | 400 | 63.04 |
| **XD (Ours)** | | 400 | **63.92 (+0.88)** |

\*: Reported DINO results from [12].

## 4.3 SACL-XD on CIFAR datasets with Tiny-sized ResNet

Table 5: CIFAR-10 linear evaluation test accuracy based on ResNet-20 trained by SACL+XD with different asymmetrical architectures.

| Method | Encoder | Linear Eval Acc. (1× model) | Teacher | Teacher Pre-trained by | Training FLOPs (e+16) |
|---|---|---|---|---|---|
| **SACL+XD (Ours)** | ResNet-20 (6×-1×) | **86.81 (+7.18)** | - | - | 8.60 |
| **SACL+XD (Ours)** | ResNet-20 (4×-1×) | **84.04 (+4.41)** | - | - | 3.90 |
| **SACL+XD (Ours)** | ResNet-20 (2×-1×) | **82.31 (+2.68)** | - | - | 0.98 |
| *SEED [14] | ResNet-20 (1 ×) | 82.86 | ResNet-20 (6×) | MoCo [7] | 180 |
| *SEED [14] | ResNet-20 (1 ×) | 81.36 | ResNet-20 (6×) | Barlow Twins [31] | 180 |
| Barlow Twins [31] | ResNet-20 (1 ×) | 79.63 | - | - | 0.25 |
| VICReg [3] | ResNet-20 (1 ×) | 79.13 | - | - | 0.25 |

\*: Re-implementation of SEED [16] on CIFAR dataset with the official code. The ResNet-20 (6×) is pretrained by 800 epochs.
†: Data augmentation and hyperparameter settings are adopted from [12]. We use [35] as the baseline of ResNet-20.

We also evaluate the performance of our method on the small-sized dataset with the tiny-sized ResNet encoders (e.g., ResNet-20 with 0.27 million parameters). We follow the data augmentation setup in [12] for the CIFAR-10 dataset. For the SACL, we sweep the asymmetry from $2\times$-$1\times$ up to $6\times$-$1\times$, the model is trained for 1000 epochs from scratch, and the results are summarized in Table 5.

With the "$6\times$-$1\times$" asymmetry, the equivalent weight sparsity is consistently held at 97.01% throughout the entire process. The linear evaluation is performed based on the slimmed $1\times$ model with 0.27 million non-zero weights. Our method achieves up to 7.18% accuracy improvements compared to the Barlow Twins baseline [35]. Compared to SEED [16], the proposed method achieves 3.95% accuracy improvements. We are aware the contrastive learning-trained large-sized ResNet-50 encoder can achieve >96% CIFAR-10 accuracy with downstream fine-tuning [18, 35], but the recent CL algorithms exhibit poor performance with tiny encoder and small-sized training samples (compared to ImageNet-1K). The superior performance of the proposed method provides valuable insights for practical self-supervised learning on limited-sized datasets with tiny-sized encoders.

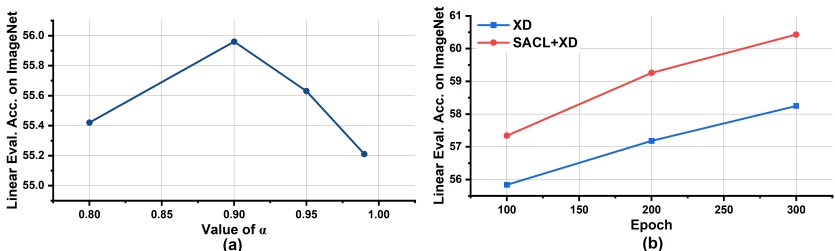

Figure 4: MobileNet-V1 ImagNet-1K accuracy vs. (a) value of $\alpha$ and (b) training epochs.

Table 6: Transfer fine-tuning of MobileNet [23, 22] pretrained by the proposed method.

| Method | Encoder | CIFAR-10 | CIFAR-100 | Aircraft | Flowers | Food-101 | Cars | Pets |
|---|---|---|---|---|---|---|---|---|
| Supervised (from scratch) | Mob-V3 (1×) | 92.97 | 73.69 | 65.37 | 79.89 | 60.30 | 68.18 | 70.97 |
| Supervised (fine-tune) | Mob-V3 (1×) | 94.53 | 78.86 | 68.29 | 89.94 | 75.84 | 82.43 | 85.87 |
| **XD (Ours)** | Mob-V3 (1×) | **94.80** | **79.00** | **71.39** | **90.05** | 75.71 | **82.77** | **89.42** |
| **SACL + XD (Ours)** | Mob-V1 (1.5×-1×) | **94.92** | **79.64** | **72.21** | **90.48** | **76.12** | **83.14** | **90.24** |
| SEED [16] | Eff-B0 (w. RN-50 teacher) | 87.5 | 63.0 | - | - | - | - | - |

Table 7: Comparison between the proposed method and other supervised high-water marks

| Model | Training Method | CIFAR-10 Acc (%) | CIFAR-100 Acc (%) | # of (remained) Param. (M) |
|---|---|---|---|---|
| ResNet-50 | Supervised Learning | 94.75 | 78.23 | 25.6 |
| ResNet-50 | Supervised + GraNet [24] | 94.64 | 77.89 | 2.6 (90% sparsity) |
| ResNet-50 | Supervised + RigL [15] | 94.45 | 76.50 | 2.6 (90% sparsity) |
| Mob-V1 | **SACL+XD (Ours) + Finetune** | **94.92** | **79.64** | 3.2 |
| Mob-V3 | **XD (Ours) + Finetune** | **94.80** | **79.00** | 3.0 |

## 4.4 Ablation Study

**The impact of $\alpha$.** We introduced the weight parameter $\alpha$ in Eq. 6 and Eq. 10 as a tunable parameter to control the importance of the inner decorrelation loss during the training process. We further evaluate the impact of the different weighting between $\mathcal{L}_{\text{SACL}}$ and $\mathcal{L}_{\text{CD}}$. We explore the impact of $\alpha$ on the ImageNet-1K dataset with the MobileNet-V1 (1×) model. The model is trained by XD only with 100 epochs from scratch. As shown in Fig. 4(a), the proposed method achieves the best performance when $\alpha$=0.9. Meanwhile, the accuracy oscillation caused by $\alpha$ is relatively stable (within ± 1%).

**The impact of training effort.** We also evaluate the performance of the proposed algorithm with different training efforts from scratch. Fig. 4(b) demonstrate the linear evaluation accuracy of MobileNet-V1 [23] on ImageNet-1K trained by different epochs. For both SACL+XD and individual XD training, the extended training effort from 100 epochs to 300 epochs leads to evident accuracy improvements. With 300 epochs of training, the proposed XD and SACL+XD method achieves 58.25% and 60.63% Top-1 linear evaluation accuracy on ImageNet-1K.

## 4.5 Transfer learning to downstream tasks

We report the transfer learning performance of the MobileNet [22, 23] encoder trained by both SACL+XD and XD. For the downstream classification, we use CIFAR-10 and CIFAR-100 as our target tasks. We also validate the pre-trained lightweight encoder on the VOC2007 dataset for downstream object detection. Follow the setup in [14], we finetune the models for 10,000 steps with SGD and batch size of 64. The experimental setup is summarized in the Appendix. Table 6 summarizes the transfer learning performance of the proposed method compared to SEED [16]. For the CIFAR tasks, our method achieves 7.42% and 16.30% accuracy improvements compared to SEED [16]. Together with the 1.83% and 5.69% improvements compared to supervised learning.

## 4.6 Comparison to SoTA energy-efficient supervised learning

In addition to the accuracy and training cost improvement compared to distillation-based contrastive learning, the powerful lightweight backbone model trained by the proposed method reveals a new perspective of energy-efficient inference compared to conventional supervised sparse training on

ResNet [24, 15]. Table 7 summarizes the CIFAR downstream comparison between the pre-trained MobileNet [29, 23] and the ResNet-50 sparsified by the recent SoTA pruning methods [24, 15]

Despite the additional fine-tuning effort, the powerful lightweight backbone pre-trained by the proposed method achieves better accuracy-model size tradeoff compared to the conventional supervised sparse learning with high element-wise sparsity. More importantly, the powerful lightweight architecture can be accelerated and deployed to the energy-constrained hardware **without** the requirement of the dedicated accelerator.

## 5 Conclusion

In this paper, we propose a novel contrastive learning (CL) algorithm designed for lightweight encoders. We first introduce the slimmed asymmetrical contrastive learning (SACL), which treats the lightweight model CL as a slimmed sparse training task with asymmetrical encoding. On top of the SACL, we propose the cross-distillation (XD) algorithm, distilling the knowledge by minimizing the decorrelation between the embeddings encoded by SACL. Compared to previous works, the proposed algorithm achieves new SOTA accuracy without introducing the large-sized teacher or expensive pre-training. Furthermore, solely training both lightweight and large-sized ResNet with the XD can still achieve superior performance. Compared to supervised learning, the ImageNet-1K-trained lightweight encoder shows superior performance in the downstream tasks with transfer learning.

**Impact and limitations.** In this work, SACL+XD works well with correlation-based loss [35] along with the shared encoder. Exploring the possibility of the proposed algorithm with the momentum-based encoder updating scheme [18, 19] could be an interesting direction. With the powerful, lightweight backbone trained by the proposed algorithm, we will further investigate the post-training quantization and hardware deployment, which further unleash the superior performance in Table 6 with practical downstream applications with actual hardware. To that end, our work can further improve the versatility of contrastive learning with energy-efficient applications (e.g., AI on edge).

**Acknowledgments.** We thank the anonymous reviewers for their comments and suggestions. This work is supported in part by the National Science Foundation under Grant No. 2144751, 2314591, 2328803, 2342726, and CoCoSys Center in JUMP 2.0, an SRC program sponsored by DARPA.

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

# A Appendix

## A.1 Algorithm

---

**Algorithm 1:** PyTorch-style pseudocode for the proposed algorithm

---

```python
# f:  encoder model
# h:  projector head
# s:  slim ratio of SACL
# slicer:  SACL slicer
# alpha:  weight between CL loss and CD loss
# lambda:  weight on the off-diagonal terms
def normalize(z):
    z_norm = (z - z.mean(dim=0)) / z.std(dim=0)
    return z_norm

for batch in trainloader:
    x_a, x_b = batch

    # SACL forward pass
    slicer.remove_mask()
    z1 = h(f(x_a))
    slicer.activate_mask()
    z2 = h(f(x_b))

    # reverse the order of input
    with torch.no_grad():
        slicer.remove_mask()
        z1t = h(f(x_b))
        slicer.activate_mask()
        z2t = h(f(x_a))

    # cross correlation
    cab = mm(normalize(z1).T, normalize(z2)) / N
    caat = mm(normalize(z1).T, normalize(z1t)) / N
    cbbt = mm(normalize(z2).T, normalize(z2t)) / N

    # Contrastive leanring loss
    cl_loss = bt_loss(cab)

    # CD loss
    dcorr_a = off_diagonal(caat).mul_(lambda).sum()
    dcorr_b = off_diagonal(caat).mul_(lambda).sum()
    cd_loss = (dcorr_a + dcorr_b) / 2

    loss = cl_loss.mul(alpha) + cd_loss.mul(1-alpha)
    loss.backward()
    optimizer.step()
```

---

## A.2 Compared to the Log-based distillation loss

From the perspective of knowledge distillation, the negative `logarithm`-based distillation loss has been widely incorporated into the "teacher-student" learning. In Section 3.2, we proposed the cross-distillation (XD) learning scheme. The distillation objective in Eq (10) is the inner decorrelation minimization between embeddings $z$ and $[\tilde{z}]$. In addition to the correlation-based distillation loss, we also investigate the `negative logarithm` (e.g, $-a \log b$) distillation loss that is employed in both supervised knowledge distillation [21] and contrastive learning [6].

To avoid the unbalanced loss magnitude, the distillation loss is introduced as the regularization term controlled by the penalty level $\gamma$:

$$\mathcal{L} = \mathcal{L}_{\texttt{SACL}}(z_A, z_B) + \gamma \mathcal{L}_{CD} \tag{11}$$
$$\mathcal{L}_{CD} = (-[\tilde{z}_A] \log z_A + -[\tilde{z}_B] \log z_B)/2 \tag{12}$$

We empirically observe that the `negative logarithm`-based distillation loss failed to outperform the proposed cross-distillation loss $\mathcal{L}_{CD}$ with inner-decorrelation minimization. As shown in the ImageNet-100 results below:

| Method | Encoder | # of Params (M) | Linear Eval Acc. (%) |
|---|---|---|---|
| **XD** | MobileNet-V1 (1×) | 3.2 | **80.30** |
| XD (w/ negative log) | MobileNet-V1 (1×) | 3.2 | 79.63* |
| Barlow Twins [35] | MobileNet-V1 (1×) | 3.2 | 78.40 |

*: Best accuracy we found with $\gamma$ =1e-3.

Although the `negative-logarithm` distillation loss is suboptimal compared to the inner decorrelation minimization, the proposed cross-distillation learning scheme is beneficial to lightweight contrastive learning, compared to the baseline [35].

## A.3 Detailed Experimental Setup of Pre-training

**ImageNet-1K** The encoders (MobileNet, EfficientNet, ResNet-50) are trained on ImageNet-1K with 100/200/300 epochs from scratch with the proposed method. We set the batch to 256 with a learning rate = 0.8. We employ the LARS optimizer with weight decay set to 1.5e-6. We set the correlation weights $\lambda$ to 0.005. The hidden layer dimension of the projector is 4096. The detailed data augmentation is summarized in Table 8

Table 8: Detailed image augmentation settings on ImageNet-1K.

| Parameter | $X_A$ | $X_B$ |
|---|---|---|
| Random crop size | 224 × 224 | 224 × 224 |
| Horizontal flip probability | 0.5 | 0.5 |
| Color jitter probability | 0.8 | 0.8 |
| Brightness adjustment probability | 0.4 | 0.4 |
| Contrast adjustment probability | 0.4 | 0.4 |
| Saturation adjustment probability | 0.2 | 0.2 |
| Hue adjustment probability | 0.1 | 0.1 |
| Gaussian blurring probability | 1.0 | 0.1 |
| Solarization probability | 0.0 | 0.2 |

**ImageNet-100** With the proposed cross-distillation method, we train the lightweight ViT model on the ImageNet-100 dataset for 400 epochs. The batch size is set to 256 with AdamW optimizer. The learning rate and weight decay are set to 0.005 and 1e-4. The detailed data augmentation is summarized in Table 9:

Table 9: Detailed image augmentation settings on ImageNet-100.

| Parameter | $X_A$ | $X_B$ |
|---|---|---|
| Random crop size | $224 \times 224$ | $224 \times 224$ |
| Horizontal flip probability | 0.5 | 0.5 |
| Color jitter probability | 0.8 | 0.8 |
| Brightness adjustment probability | 0.4 | 0.4 |
| Contrast adjustment probability | 0.4 | 0.4 |
| Saturation adjustment probability | 0.0 | 0.2 |
| Hue adjustment probability | 0.1 | 0.1 |
| Gaussian blurring probability | 1.0 | 0.1 |
| Solarization probability | 0.0 | 0.2 |

**CIFAR-10** The proposed method is trained from scratch by 1,000 epochs with LARS-SGD opti-mizer [33]. We use 256 batch size along with 0.3 learning rate and $1e - 4$ weight decay. The Cosine learning rate scheduler is used with 10 epochs of warmup training. The detailed data augmentation is summarized in Table 10.

Table 10: Detailed image augmentation settings on CIFAR-10.

| Parameter | $X_A$ | $X_B$ |
|---|---|---|
| Random crop size | $32 \times 32$ | $32 \times 32$ |
| Horizontal flip probability | 0.5 | 0.5 |
| Color jitter probability | 0.8 | 0.8 |
| Brightness adjustment probability | 0.4 | 0.4 |
| Contrast adjustment probability | 0.4 | 0.4 |
| Saturation adjustment probability | 0.2 | 0.2 |
| Hue adjustment probability | 0.1 | 0.1 |
| Gaussian blurring probability | 0.0 | 0.0 |
| Solarization probability | 0.0 | 0.2 |

## A.4 Detailed Experimental Setup of Downstream Fine-tuning

We evaluate the transferability of the pre-trained lightweight model on downstream tasks, including CIFAR-10, CIFAR-100, and VOC2007. Following the settings in [14], we fine-tuned the models for 10,000 steps with SGD and batch size of 64. The learning rate is set to 0.1 with no weight decay. The input samples are resized to $224 \times 224$ to maintain the dimensionality as the pre-trained model. The checkpoint of the pre-trained lightweight model will be released soon.

## A.5  Detailed Training Cost of the Proposed Method

In addition to the FLOPs comparison in Table 2, we evaluate the training cost of the proposed method based on the actual training time per epoch, together with the total activation count (feature map pixels) for each forward pass of the training process, as shown in Table 11 and Table 12, respectively.

Table 11: Training time comparison between the proposed method and the distillation-based CL

| Model | Training Method | Teacher | Training time / epoch | GPU Type | Batch Size |
|---|---|---|---|---|---|
| SEED [16] | MobileNet-V3 | ResNet-50 | 35 min 20 sec | A100 (80G) | 256 |
| **SACL-XD (Ours)** | MobileNet-V3 1.5×-1× | N/A | 26 min 02 sec | A100 (80G) | 256 |
| **XD Only (Ours)** | MobileNet-V3 | N/A | 16 min 15 sec | A100 (80G) | 256 |

Table 12: Activation count comparison between the proposed method and the distillation-based CL

| Method | Encoder | Teacher | Act. Count (E+07) | ImageNet-1K Accuracy (%) |
|---|---|---|---|---|
| **SACL+XD (Ours)** | Eff-B0 (1.5×-1×) | N/A | 1.54 | **65.32** |
| **XD Only (Ours)** | Mob-V3 (1×) | N/A | **0.90** | **59.34** |
| SSL-Small [27] | Mob-V3 | N/A | 0.90 | 48.70 |
| SSL-Small [27] | Eff-B0 | N/A | 0.68 | 55.90 |
| ReKD [36] | Mob-V3 | ResNet-101 | 2.03 | 59.60 |
| ReKD [36] | Mob-V3 | ResNet-50 | 1.53 | 56.70 |
| ReKD [36] | Eff-B0 | ResNet-50 | 1.75 | 63.40 |
| SEED [16] | Mob-V3 | ResNet-50 | 1.52 | 55.20 |
| SEED [16] | Eff-B0 | ResNet-101 | 2.26 | 61.30 |

