# OpenReview forum: "Slimmed Asymmetrical Contrastive Learning and Cross Distillation for Lightweight Model Training"
_NeurIPS.cc/2023/Conference — NeurIPS 2023 poster_

### Official Review · Reviewer_8Hzi · 2023-07-03

**Soundness:** 3 good
**Presentation:** 3 good
**Contribution:** 2 fair
**Rating:** 6
**Confidence:** 4

**Summary:**

This paper proposed a new method to enable effective contrastive learning (CL) on the lightweight encoder without a mega-size teacher model, which can thus reduce the training cost. More specifically, it expanded the target model to a larger model, which shares its weights with the target model. Then the CL problem is formulated as slimmed training task with asymmetrical encoding. Furthermore, this work incorporated cross-distillation to further minimize the decorrelation between the embeddings of the same view but from different encoders. Experimental results show that the proposed method can outperform the existing baselines with much smaller computational costs on the lightweight models.

**Strengths:**

1. The paper is well-motivated. Figure 1 clearly shows that the efficient CL of the lightweight model is challenging and what improvement the proposed method can make to this problem.

2. The proposed method is reasonable and Figure 2 clearly shows the key idea.

3. The proposed method can not only be effective in the CL of a lightweight model but also in a mega-size model.

4. It is good to see that the authors include the experimental results to show that the trained encoder can be well generalized to different types of downstream tasks.

**Weaknesses:**

1. Although the authors claim Ref [25] cannot work on the mega-size encoder, it should be compared with the proposed method on the lightweight encoder.

2. It would be better if the authors could show the performance of supervised learning in the tables of the experiment section to serve as the upper bound of the proposed method.

3. It would be better if the authors can add a column for the training cost in Table 3, 4 and 5.

4. Some technical details are not clear, which is shown in the question part.


**Questions:**

1. Can you provide some explanation about why the proposed method can achieve better performance than the supervised learning in Table 6?

2. In Figure 1, there are two red spots annotated with "SCAL + XD". What are the differences?

3. Can the authors provide an example to illustrate the meaning of the style of "K $\times$ -1 $\times$ " for the slimming ratio? For instance, if the pruning ratio of the task model is 0.5 compared with the expanded model,  what is the value of K and $s$?

4. In Equation (7), what is the meaning of $n, i, j$?

5. For $h_{\phi}$ from the two different encoders, are they of the same dimension? Do $Z_{A}$ and $Z_{B}$ have the same dimension?



**Limitations:**

Although the author claims that their method does not need an extra mega-size teacher, their method still needs an expanded version of the task model for effective CL.

---

> ### Author Rebuttal · Authors · 2023-08-10
>
> ## Response to Reviewer 8Hzi
>
> Thank you for your thorough review on our paper and we appreciate your positive feedback.
>
> ### Weakness
>
> **Weakness 1:** Although the authors claim Ref [25] cannot work on the mega-size encoder, it should be compared with the proposed method on the lightweight encoder.
>
> **Response to Weakness 1:** Thank you for your feedback. [25] proposed an input-scaling and cropping strategy dedicated to lightweight encoders. Noticeably, [25] introduces **four** different types of multi-view sampling parameters as the hyperparameters of the training process. Such a fine-grained scaling strategy is orthogonal to our proposed method, which focuses on a generic training solution for lightweight contrastive learning with normal initialization and commonly-used data augmentation strategy as prior contrastive learning methods. Therefore, our method has better generality compared to [25], for both lightweight models (SACL+XD) and large-sized encoders (XD only). The model-specific data augmentation and local-view scaling strategy will remain as our future work.
>
> ---
>
> **Weakness 2:** It would be better if the authors could show the performance of supervised learning in the tables of the experiment section to serve as the upper bound of the proposed method.
>
> **Response to Weakness 2:** Thank you for pointing this out. Due to the page limit of rebuttal,  we will include the supervised learning results as the upper bound of the performance of all the tables in the next version of the paper.
>
> ---
>
> **Weakness 3:** It would be better if the authors can add a column for the training cost in Table 3, 4 and 5.
>
> **Response to Weakness 3:** Thank you for highlighting this. We updated Table 5 with the total training FLOPs as **Table 7** of the attached PDF. Due to the page limit of rebuttal, we will include the detailed training FLOPs in Table 3 and 4 in the next version of the paper.
>
> ------
>
> ### Questions:
>
> **Q1:** Can you provide some explanation about why the proposed method can achieve better performance than the supervised learning in Table 6?
>
> **A1:** Thank you for your question. As presented in the prior works of contrastive learning, aligning the latent information leads to enhanced robustness in learning various visual representations.
> Compared to supervised learning with deterministic labels, the encoder (e.g., ResNet-50) trained by contrastive learning can achieve better performance in the downstream small vision tasks with minimum tuning, as shown in [16, 31].
>
> ---
>
> The proposed cross-distillation algorithm not only maximizes the correlation between latent features but also avoids feature aliasing across different dimensionality, which further enhances the representation learning of the model.
>
> **Q2:** In Figure 1, there are two red spots annotated with "SCAL + XD". What are the differences?
>
> **A2:** Thank you for pointing this out and we apologize for the confusion. The large dots and small dots in Figure 1 represent the EfficientNet-B0 encoder (4.0 Million parameters) and MobileNet-V3 encoder (3.0 Million parameters), respectively, as shown in the bottom right legend of the figure. We will further clarify that in the next version of the paper.
>
> ---
>
> **Q3:** Can the authors provide an example to illustrate the meaning of the style of "K -1 " for the slimming ratio? For instance, if the pruning ratio of the task model is 0.5 compared with the expanded model, what is the value of K and s?
>
> **A3:** Thank you for your question. By saying "K-1" asymmetry, it means that we slim both input and output channels by 1/K times. If $K$=2, the input/output channels are slimmed by half, which leads to $s=75\%$ structured sparsity.
>
> ---
>
> **Q4:** In Equation (7), what is the meaning of $n$, $i$, $j$?
>
> **A4:** Thank you for your question. $n$ represent the index of batch, and $i$ and $j$ represent the dimensionality indices across the latent output. Correspondingly, $C_{i,j}$ represents the ${i,j}$ element of the correlation matrix. We will further clarify the symbols in the next version of the paper.
>
> ---
>
> **Q5:** For from the two different encoders, are they of the same dimension? Do $Z^A$ and $Z^B$ have the same dimension?
>
> **A5:** To make the dimensionality matched and non-empty at the output of the last layer, we skip the slimming in the output channel (input channel is still slimmed) of the final layer. Therefore, $Z^A$ and $Z^B$ will have the same dimensionality.

---

> > ### Comment · Reviewer_8Hzi · 2023-08-12
> > **Confirmation of Reading the Rebuttal**
> >
> > Thanks for the detailed feedback from the authors. The answers they provided addressed most of my questions and helped me to have a deeper understanding of the contribution and techniques of this work. Therefore, I prefer to keep my previous score to show my support for this work.

---

### Official Review · Reviewer_J8Hb · 2023-07-04

**Soundness:** 3 good
**Presentation:** 3 good
**Contribution:** 3 good
**Rating:** 6
**Confidence:** 4

**Summary:**

This paper proposes a self-supervised contrastive learning method to improve the light-weight network performance and reduce the training cost. It mainly consists of two components: slimmed asymmetrical contrastive learning (SACL) and cross-distillation (XD), which are able to train the efficient network from scratch without the usage of pre-trained strong teacher. Experiments the effectiveness of the proposed method in terms of accuracy and training FLOPs reduction.

**Strengths:**

1. SACL+XD is effective to improve the light-weight network performance with efficient training.
2. well-written and easy to follow this paper.


**Weaknesses:**

1. The SACL+XD seems to apply the slimmable neural network[ref-1] to Disco[15]. Therefore, the novelty seems to be limited.

2. [Experimental issues.] (1)This paper mainly uses several efficient backbones (e.g. MobileNets) as the base model backbone, It lacks of comparisons on the more dense backbones, such as ResNet-50. (2) SACL+XD trains the models from scratch by removing a unified amount of channels based on the lowest magnitude score. I think the initial weights should have an effect on the final performance, as the channel selection in a slimmed network f_{\theta}^s may be different under the different weight initialization settings. However, the experiments lack such comparison about the initialization, as well as the mean accuracy with std. (3) In this paper, computation reduction is evaluated by training FLOPs. The authors better add the practical training time for a more comprehensive comparison.


[Ref-1] Yu J, Yang L, Xu N, et al. Slimmable neural networks[J]. arXiv preprint arXiv:1812.08928, 2018.

**Questions:**

Refer to the Weaknesses.

**Limitations:**

Yes. The authors well address the impact and limitations.

---

> ### Author Rebuttal · Authors · 2023-08-10
>
> ## Response to Reviewer J8Hb
>
> Thank you for your thorough review on our paper and we appreciate your positive feedback.
>
> **Weakness 1:** The SACL+XD seems to apply the slimmable neural network[ref-1] to Disco[15]. Therefore, the novelty seems to be limited.
>
> **Response to Weakness 1:** Thank you for your feedback, but we respectfully disagree with your comments.
>
> Firstly, the slimmable network [ref-1] is designed for training **multiple** sparse subset networks with **supervised learning**. However, the proposed method introduced two novel techniques of (1) *slimmed asymmetry* and (2) *cross distillation* for **self-supervised** contrastive learning. Our paper aims to resolve the low accuracy issue of lightweight models (e.g., MobileNet) trained by contrastive learning schemes. The objective is to obtain a **single** powerful, lightweight model via contrastive learning without introducing the large-sized teacher, which can also achieve superior performance in the downstream vision tasks, as summarized in Table 2 and Table 6 of the original manuscript.
>
> Secondly, the proposed method has key essential differences  and novelties compared to Disco [15]. In particular, Disco [15] introduced an auxiliary "mean student'' encoder in addition to the student-teacher pair. In other words, Disco [15] complicates the lightweight model contrastive learning with complex distillation design, and the large-sized ResNet teacher model **is still preserved** in the contrastive learning process. On the contrary, our proposed method **completely eliminates** the "ResNet teacher" distillation and teacher pre-training for lightweight contrastive learning, while achieving superior performance in both ImageNet and downstream tasks.
>
> In conclusion, the proposed method is orthogonal to slimmable neural networks [ref-1] and Disco [15]. Furthermore, our method completely eliminates the "student-teacher'' distillation that is heavily exploited in Disco [15].
>
> ---
>
> **Weakness 2-1:** (1) This paper mainly uses several efficient backbones (e.g. MobileNets) as the base model backbone, It lacks of comparisons on the more dense backbones, such as ResNet-50.
>
> **Response to Weakness 2-1:** While it is correct that the main objective of our paper is to train the high-performance, lightweight encoder via contrastive learning (which is a long-lasting problem in prior contrastive learning methods), our original manuscript still also validated the proposed cross-distillation (XD) method on the large-sized ResNet-50 backbone model. To that end, we respectfully disagree with your comment that it lacks comparisons on denser backbones like ResNet-50. As shown in Table 2 of the original manuscript, our method can achieve improved performance on ResNet-50 compared to the recent contrastive learning methods, which proves the versatility of our proposed method. In conclusion, the proposed method achieves good performance with **both** lightweight and large-sized encoder models.
>
> **Weakness 2-2:** (2) SACL+XD trains the models from scratch by removing a unified amount of channels based on the lowest magnitude score. I think the initial weights should have an effect on the final performance, as the channel selection in a slimmed network $f_{\theta}^s$ may be different under the different weight initialization settings. However, the experiments lack such comparison about the initialization, as well as the mean accuracy with std.
>
> **Response to Weakness 2-2:** Thank you for your question. We want to highlight the fact the major focus of our method is providing a generic and versatile contrastive learning solution for lightweight models.
> In particular, our motivation is directly training the lightweight encoder via contrastive learning without introducing expensive distillation [9, 14, 15, 32], additional tuning on initialization [24], or fine-grained view scaling and sampling [25]. With the simple training setup of the proposed method, the learned visual representation via our proposed method leads to superior performance on the downstream tasks compared to the lightweight model trained by supervised learning.
> Unlike previous work [24], we posit that initializing the weights with minimal distillation may have minimal impact on our asymmetrical slimming. This is because the lightweight encoder is sliced from the widened model, and this corrupts the benefits of the weight initialization.
>
> ---
>
> **Weakness 2-3:** (3) In this paper, computation reduction is evaluated by training FLOPs. The authors better add the practical training time for a more comprehensive comparison.
>
> Thank you for pointing this out. To address your question, we summarize the comparison of training time per epoch in **Table 4** of the attached PDF.
>
> These are measured by running a single training task on a single Nvidia A100 GPU. We will include this table in the next version of the paper together with all the other models.

---

> > ### Comment · Reviewer_J8Hb · 2023-08-17
> > **After the rebuttal**
> >
> > Thanks for the authors' response. My concerns are well addressed in the rebuttal, such as the novelty problem and several experiments. I will increase my score to weak accept this paper.

---

### Official Review · Reviewer_UkVS · 2023-07-06

**Soundness:** 4 excellent
**Presentation:** 4 excellent
**Contribution:** 3 good
**Rating:** 6
**Confidence:** 4

**Summary:**

This paper introduces a self-supervised contrastive learning algorithm designed specifically for training lightweight models, eliminating the need for a large teacher model. The algorithm comprises two main components: slimmed asymmetrical contrastive learning (SACL) and cross-distillation (XD). The authors evaluate their approach using different lightweight models and datasets, demonstrating its superiority in terms of performance and efficiency over existing methods.

**Strengths:**

1, This paper is well-motivated. This paper tackles a significant and practical challenge of training lightweight models using contrastive learning, which has received limited attention in previous research efforts.
2, The paper presents a novel and straightforward concept of slimming the host encoder, creating asymmetrical encoding paths for contrastive learning. This innovative approach effectively reduces training costs and enhances the performance of lightweight models.
3, The writing is clear and easy to follow.
4, The paper conducts thorough experiments on diverse models, datasets, and downstream tasks, and conducts comprehensive comparisons with state-of-the-art methods. The results strongly validate the efficacy and general applicability of the proposed method.


**Weaknesses:**

1, To enhance the clarity and understanding of the proposed methods, it would be beneficial to provide additional intuition. Specifically, more insights can be provided on how cross-distillation aids in overcoming the distortion resulting from asymmetrical encoding. Additionally, exploring the trade-offs associated with different levels of asymmetry and sparsity would further enrich the paper.



**Questions:**

1, While the paper focuses on the limitations of contrastive learning (CL) for lightweight models, it is worth considering whether alternative self-supervised learning methods, such as generative pre-training or reconstruction-based pre-training, are effective for these models. Therefore, it would be valuable to include a comparison between the proposed methods and these alternative approaches to provide a comprehensive understanding of the performance of different self-supervised learning techniques on lightweight models.

**Limitations:**

Please refer to the weakness and questions.

---

> ### Author Rebuttal · Authors · 2023-08-10
>
> ## Response to Reviewer UkVS
>
> ### Weaknesses
>
> Thank you for your thorough review on our paper and we appreciate your positive feedback.
>
> **Weakness 1:** To enhance the clarity and understanding of the proposed methods, it would be beneficial to provide additional intuition. Specifically, more insights can be provided on how cross-distillation aids in overcoming the distortion resulting from asymmetrical encoding. Additionally, exploring the trade-offs associated with different levels of asymmetry and sparsity would further enrich the paper.
>
> **Response to Weakness 1**: Thank you for your comments. The asymmetric encoding of the proposed SACL algorithm introduces the distorted latent information in one branch of contrastive encoding. Meanwhile, the asymmetrical encoding + contrasive learning can also be treated as a distillation task. However, na\"ively minimizing the distance between the dense and slimmed encoders is not the perfect solution for learning, as pointed out in Section 1.2 of the supplementary material.
>
> Therefore, the design of the contrastive loss should (1) align the features across different dimensions of the latent space and (2) avoid the feature mismatch and collapse between the slimmed and original features. As a result, the cross-distillation loss is added as a moving average term on top of the original SACL loss function.
>
> Regarding the tradeoffs with different sparsity levels, we explored the different asymmetry-accuracy tradeoffs in Table 5 of the original manuscript. To further clarify the tradeoff between performance-accuracy-training cost, we extended Table 5 of the original manuscript with training FLOPs and summarized the comparison in **Table** 7 of the attached PDF.
>
> ------
>
> ### Questions
>
> **Q1:** While the paper focuses on the limitations of contrastive learning (CL) for lightweight models, it is worth considering whether alternative self-supervised learning methods, such as generative pre-training or reconstruction-based pre-training, are effective for these models. Therefore, it would be valuable to include a comparison between the proposed methods and these alternative approaches to provide a comprehensive understanding of the performance of different self-supervised learning techniques on lightweight models.
>
> **A1:** Thank you for pointing this out. Currently, our paper mainly focuses on contrastive learning with lightweight models. We will further investigate the lightweight generative pre-training (GPT) in our future work.

---

### Official Review · Reviewer_CDjk · 2023-07-11

**Soundness:** 3 good
**Presentation:** 3 good
**Contribution:** 3 good
**Rating:** 6
**Confidence:** 3

**Summary:**

The authors propose a combination of two method: Slimmed Assymterical Contrastive Learning(SACL) , and Cross-Distilation(XD) with a correlation-maximization loss.  SACL does magnitude pruning of filters at each epoch, and the pruned model is used as a encoder of one of two views used in the contrastive loss.  They find that XD with an efficient network already provides competitive performance, and SACL brings performance to state-of-the-art. Evaluations are done with linear evaluation on ImageNet, CIFAR, and VOC2007.

**Strengths:**

Originality: Though cross-distillation and training sparse networks are both known techniques, their combination with correlation-based contrastive learning is a novel combination.

Quality: The quality of the experimental design is sufficient to be convincing, and the idea makes sense, and seems easy to implement, increasing potential impact.

Clarity: The clarity of the writing is reasonably high, with the exception of the description of XD(see weakneses).

Significance: Self-supervised pre-training has proven to be quite useful, and efforts to remove key limitations in encoder size can achieve important practical impact.

**Weaknesses:**

* Experiments: One key advanatage of SSL methods  is in transfer learning, and learning general features. However, the transfer learning study in this work is relatively small, with only CIFAR and VOC presented. In my mind, this is the biggest weakness of the work.
*  It's not explicitly stated how XD experiments are setup. Are they cross-distillation across two instances of a network of the same architecture (but independent weights), like Figure 2c but without slimming?
* Slimming idea is general, but implementation seems CNN-specific.
* Line 85: Minor, but saying ResNet-50 is mega-sized is a bit of an overclaim IMO.
* Lack of substantial discussion of different axes of efficiency (FLOPs, wall time, activation count). Each of these metrics is useful in its own way, but training FLOPS is primary focus in this work.

**Questions:**

* How does the method compared to a supervised high-water mark, beyond the experiments in Table 6?
* How is transfer performance, beyond CIFAR and VOC?
* How could one implement an equivalent slimming procedure for ViTs? Having a plausible path for implementation in ViT's could further strengthen the work.
* In Table 2, row SSL-Small, why is the training flops much larger than XD-only? It is same architecture, same epoch count, no pretraining.
*

**Limitations:**

Limitations are adequately addressed?

---

> ### Author Rebuttal · Authors · 2023-08-10
>
> ## Rebuttal to Reviewer CDjk
>
> Thank you for your thorough review on our paper and we appreciate your positive feedback.
>
> ### Weaknesses
>
> **Response to Weakness 1**: Thank you for pointing this out. To address your comments, we extended the experiments in Table 6 of the original manuscript with extended datasets and results, as shown in **Table 1** of the attached PDF file.
>
> As shown in Table 1, the lightweight model trained by the proposed method can be powerful across a wide range of %spectrums in terms of downstream tasks. Compared to the model that is pre-trained on ImageNet-1K with **supervised learning**, the proposed method consistently achieves superior performance among all tasks, which indicates the high effectiveness of the proposed method in visual representation learning. We will include this updated table in the next revised version of the paper.
>
> ------
>
> **Response to Weakness 2:** Thank you for your comments. We want to clarify the settings of XD as follows:
>
> - The cross distillation (XD) and SACL-XD exhibit the same settings where the weights are **shared** between two encoders. Since the augmentation and sparsity are collectively applied to the encoder, the latent output needs to preserve the similarity by having identical non-slimmed weights.
> - As described from line 230 to line 241 of the original manuscript, minimizing the cross distillation loss $\mathcal{L}_{\texttt{CD}}$ avoids the aliasing features across different dimensions of the augmented input. Therefore, consistent encoding is required, which necessitates the shared weights between two encoders.
>
> ------
>
> **Response to Weakness 3:** We will modify the "mega-sized" encoder to "large-sized" encoder in the next revised version of the paper.
>
> ------
>
> **Response to Weakness 4:** Thank you for your feedback on this. To address your question, we extended Table 5 of the original manuscript with the results of the ViT-Tiny [12] model trained by the proposed SACL-XD method under the slimming ratio of 1.25$\times$ - 1$\times$. The updated table (**Table 2**) is attached in the PDF file.
>
> Specifically, we slim down the embedding dimensionality of the ViT model, which further leads to the reduced model width in both input and output features of the fully-connected layers (and normalization layers). As shown in Table 2, the proposed SACL+XD continuously achieves improved accuracy compared to the prior baselines.
>
> We will include the details of the transformer-based SACL-XD in the next version of the paper.
>
> ------
>
> **Response to Weakness 5:** Thank you for pointing this out. To address your comment, we summarized the forward pass activation count of the training process in Table 3 of the attached document.
> Specifically, we compute the total activation count in **both** contrastive encoders (or teacher and student) in each forward pass during the training process. For the previous distillation process, the intermediate activation resulted from both student and large-sized teacher models. As shown in **Table 3**, the proposed method trains the lightweight encoder without introducing the large ResNet teacher model, collectively achieving better accuracy and lower activation count, which will further lead to reduced memory consumption during the training process.
>
> In **Table 4** of the attached PDF file, we also summarize the comparison of training time per epoch. These are measured by running a single training task on a single Nvidia A100 GPU.
>
> ------
>
> ### Questions
>
> **Q1:** How does the method compare to a supervised high-water mark, beyond the experiments in Table 6?
>
> **A1:** Thank you for your question. Regarding the general comparison with the supervised learning on the small-sized vision tasks (e.g., CIFAR-10/100), we validated the proposed algorithm from two perspectives: 1) Comparison with the large-sized dense model and 2) The sparse model pruned from the large-sized dense model. We use the widely-reported CIFAR performance as an example and summarize the comparison in **Table 5** of the attached PDF file.
>
> [Ref-1]: U.Evci, et al., "Rigging the lottery: Making all tickets winners", ICML, 2020
>
> [Ref-2]: S.Liu, et al., "Sparse Training via Boosting Pruning Plasticity with Neuroregeneration", ICLR 2021
>
> As shown in the table, the lightweight MobileNet-V3 model trained by the proposed method achieves better performance compared to the SoTA sparse training algorithms with similar parameter sizes ($\sim$3 million). In other words, the proposed method can successfully train a strong, lightweight vision learner with superior downstream performance **without** compression or pruning.
>
> ------
>
> **Answer to Q2:** Please refer to our response to Weakness 1.
>
> ------
>
> **Answer to Q3:** Please refer to our response to Weakness 4.
>
> ------
>
> **Q4:** In Table 2, row SSL-Small, why is the training flops much larger than XD-only? It is same architecture, same epoch count, no pretraining.
>
> **A4**: Thank you for pointing this out.
> The FLOPs value for SSL-Small [24] in Table 2 of the original manuscript was incorrect due to a mistake, which we apologize for.
>
> We corrected the top part of Table 2 of the original manuscript as Table 6 in the attached PDF file.
> As reported in Table 7 and Table 10 of [24], the model is trained by 800 epochs with contrastive learning and initialized with 2 epochs SEED pre-training.  Specifically, we adopt the Thop as the FLOPs counter. For EfficientNetB0, the total training FLOPs **per epoch** is 4.2E+15, and the total FLOPs of 800 epochs is 3.4E+18. Together with the 2 epochs pertaining with SEED [14] (5.29E+16 per epoch), the total training FLOPs of SSL-Small = 3.43E+18. Applying the same computation to MobileNet-V3 leads to the total FLOPs = 1.94E+18.
>
> Due to the larger training effort and pre-training, the training cost of [24] is still more expensive than ours.

---

> > ### Comment · Reviewer_CDjk · 2023-08-10
> > **Acknowledgment**
> >
> > Thank you to the authors for their rebuttal. I have kept my score but am increasing my confidence to reflect my now stronger understanding of the paper.

---

### Author Rebuttal · Authors · 2023-08-10

Dear Reviewers and AC,

We thank all the reviewers for their helpful feedback, comments, and questions. As mentioned in the individual rebuttal thread, we upload the 1-page PDF attachment that includes all the updated tables for your reference.

Please let us know if you have any further questions or comments.

Thank you.

---

### Decision · Program_Chairs · 2023-09-21

**Decision:**

Accept (poster)

**Comment:**

This paper proposes a self-supervised learning method for training lightweight models, without the help of the large teacher model. The model contains two main components: slimmed asymmetrical contrastive learning (SACL) and cross-distillation (XD). The paper demonstrates the effectiveness of the proposed algorithm on different lightweight baseline models and datasets. The paper receives four weak accepts. The rebuttal and response well resolve the reviewers' concerns. The AC recommends acceptance for this paper.